# Surveillance of SARS-CoV-2 in Sewage Treatment Plants between January 2020 and July 2021 in Taiwan

**DOI:** 10.3390/pathogens10121611

**Published:** 2021-12-10

**Authors:** Wei-Lun Huang, Wen-Bin Fann, Rong-Jun Shen, Yi Chu, Jyh-Yuan Yang

**Affiliations:** Center for Diagnostics and Vaccine Development, Centers for Disease Control, Ministry of Health and Welfare, Taipei 115210, Taiwan; hwl@cdc.gov.tw (W.-L.H.); fanndino@cdc.gov.tw (W.-B.F.); rongjun@cdc.gov.tw (R.-J.S.); chu@cdc.gov.tw (Y.C.)

**Keywords:** SARS-CoV-2, sewage treatment plants, surveillance, Taiwan

## Abstract

An outbreak of a new type of coronavirus pneumonia (COVID-19) began in Wuhan, Hubei Province, China, at the end of 2019, and it later spread to other areas of China and around the world. Taiwan reported the first confirmed case from an individual who returned from Wuhan, China, in January 2020 for Chinese New Year. Monitoring microbes in environmental sewage is an important epidemiological indicator, especially for pathogens that can be shed in feces such as poliovirus. We have conducted additional SARS-CoV-2 sewage testing since January 2020 using a well-established poliovirus environmental sewage surveillance system in Taiwan. Wastewater samples were collected from 11 sewage treatment plants from different parts of Taiwan twice a month for laboratory testing. By the end of July 2021, 397 wastewater specimens had been tested, and two samples were positive for SARS-CoV-2. These two wastewater samples were collected in the northern region of Taiwan from Taipei (site A) and New Taipei City (site C) at the beginning of June 2021. This result is consistent with the significant increase in confirmed COVID-19 cases observed in the same period of time. As the pandemic ebbed after June, the wastewater samples in these areas also tested negative for SARS-CoV-2 in July 2021.

## 1. Introduction

In December 2019, many cases of unidentified viral pneumonia occurred in Wuhan City, Hubei Province, Mainland China. On 12 January 2020, the World Health Organization (WHO) named this virus “2019 novel coronavirus (2019-nCoV)” and announced that this was a Public Health Emergency of International Concern (PHEIC) on 30 January 2020. This virus was officially named “severe acute respiratory syndrome coronavirus 2 (SARS-CoV-2)” by the International Society of Virology on 11 February 2020, and the WHO also officially named the disease caused by this virus “coronavirus disease 2019, COVID-19”. On 15 January 2020, Taiwan announced the newly added “severe special infectious pneumonia” as the fifth category of statutory infectious disease, and it is also known as “new coronary pneumonia”. As of 24 September 2021, a total of 194 countries around the world have reported cases of COVID-19, and a total of approximately 229.61 million confirmed cases have been reported globally, among which, 4,725,685 people have died. The global fatality rate is 2.06%. The top five countries in terms of the cumulative number of confirmed diagnoses in the world are the United States (18%), India (15%), Brazil (9%), the United Kingdom (3%), and Russia (3%) [1].

The primary routes of viral transmission (SARS-CoV-2) were considered to be through droplet infections and person-to-person close contact, but later, an increasing possibility of fecal–oral transmission was evident from various published studies, especially in neonatal cases [2,3,4,5]. In the context of the ongoing COVID-19 pandemic, SARS-CoV-2 shed into wastewater from the upper gastrointestinal and upper respiratory systems and via feces may be detected. This shows the probability of environmentally mediated transmission. As the source of transmission of SARS-CoV-2 is still unknown, the wastewater transmission pathway could become an important mode [6]. Hence, the presence of SARS-CoV-2 in contaminated wastewater samples and its role in transmission need to be investigated.

A previous study investigating a total of 4243 COVID-19 patients in Hong Kong reported that 17.6% of the patients exhibited gastrointestinal symptoms, and SARS-CoV-2 RNA was detected in stool samples from a higher proportion (48.1%) of the patients, even in stool collected after respiratory samples had negative results [7]. Furthermore, some studies have reported the presence of viral RNA in the stools of COVID-19 patients in percentages ranging from 16.5 to 100% [8,9,10,11]. SARS-CoV-2 is highly stable at 4 °C. As the incubation temperature increases to 70 °C, the virus is inactivated after approximately 5 min. In addition, coronaviruses are generally known to be susceptible to chlorine and inactivated relatively faster in water than nonenveloped viruses. Chlorine-based disinfectants, such as household bleach, chloroxylenol, chlorhexidine, and benzalkonium chloride, have been found to be effective for inactivating SARS-CoV-2 [12].

An environmental surveillance system was established to monitor the circulation of wild and vaccine-related poliovirus in Taiwan since July 2012 [13]. Specimens of untreated sewage were collected in northern, central, southern, and eastern Taiwan from 11 (sites A to M) representative sewage treatment plants (Figure 1). In response to the international threat of wild-type poliovirus (WPV) importation and changes to the national vaccination policy, we adopted the WHO guidelines for the environmental surveillance of circulation in Taiwan [14]. Two-phase dextran 40/polyethylene glycol (PEG) separation and a cell culture were performed to monitor environmental viral circulation. In addition to the environmental monitoring of poliovirus, we conducted environmental monitoring for enterovirus (EV), coxsackievirus A (CVA), coxsackievirus B (CVB), echovirus (Echo), human rhinovirus (HRV), adenoviruses (AdV) and mammalian orthoreovirus (MRV) every month. Through this environmental surveillance, HAV was also detected in wastewater, accompanied by hundreds of HAV (+) patients in Taiwan from our study [15].

Since the global spread of the COVID-19 pandemic, Taiwan reported the first overseas immigration and locally confirmed case in January 2020 (Figure 2). As of 24 September 2021, a total of 16,168 people have been diagnosed, among which 841 have died, with a fatality rate of 5.2%. Analyzed by month, the initial number of confirmed cases was 203 in March 2020. In addition, from January 2020 to April 2021, the number of confirmed cases per month was between 98 and 124. From 1 May to 14 May 2021, the confirmed number of cases per day was between 1 and 32. On 15 May 2021, the confirmed number surged to 186 due to an outbreak in the Taipei area and this rapidly spread to other parts of Taiwan. From 16 May to 31 May 2021, the number of new cases per day exceeded 200. The number of cases per day reached a maximum of 722. The total number of confirmed cases reached 7372 in May, which is also the highest number in a single month since the first case of COVID-19 occurred in Taiwan. After that, the number of cases showed a downward trend, with 6298 cases in June, 888 cases in July, 315 cases in August and 215 cases in September. The pandemic has been brought under control after adopting serial preventive measures, such as wearing masks, real-name registration, expanding vaccination, expanding screening, and quarantine [1].

Due to the spread of the COVID-19 pandemic, starting in January 2020, we used the poliovirus environmental sewage monitoring mechanism that has been established in Taiwan to conduct additional new coronavirus monitoring. We hope that through the collection and testing of environmental wastewater, we can actively discover the new coronavirus. To date, more than thirty countries [16] have reported research on SARS-CoV-2 in wastewater but no study has provided insights into the presence of SARS-CoV-2 in wastewater in Taiwan. This is the first study reporting the detection of SARS-CoV-2 RNA in wastewater in Taiwan.

## 2. Results

According to the original plan, the sampling frequency was once every two weeks, but as Chinese New Year took place in January 2020 and February 2021, sampling was only performed once in these two months. Due to the small-scale nosocomial infection clustering of COVID-19 in Taoyuan (site M) at the end of 2020, an extra sampling collection time point was added at site M in January 2021 for the enhanced monitoring of SARS-CoV-2 in the Taoyuan area (Figure 1 and Table 1).

Among the sewage samples, two positive SARS-CoV-2 results were found on 1 June 2021 from two northern regions (site A and site C) in Taiwan, Ct 41.1 and 41.8, respectively. We repeated these two samples and obtained consistent results. In addition, these two results were further confirmed by another automatic system, an Abbott Alinity m SARS-CoV-2 assay targeting different genes (RdRp and N), which is also approved by the FDA EUA and the CE IVD. These two positive samples also obtained the same results using Alinity m, confirming the results as true positive SARS-CoV-2 results (data not shown). The remaining nine collection sites were all negative in June. Then, the results of the second sampling in June and two samplings in July were all negative. From January 2021, a total of 397 samples were collected: two samples were positive, and the positive rate was 0.5%.

Compared with the SARS-CoV-2 results, the use of sewage samples for routine environmental surveillance systems mainly detects poliovirus, EV, CVA, CVB, Echo, HRV, AdV, MRV, and other viruses. From January 2010 to July 2021, no poliovirus was detected. EVs were detected, except for in June, July, September, November, and December 2020, with the largest number of EVs being detected in February 2020. In 2021, EV was detected in January, March, and April, and no EV was detected in the remaining months. During this period, the EV positivity rate was 5.3%. CVA was detected in January, February, March, May, August, and September 2020 and was not detected in the remaining months of 2020 and 2021. The positive rate of CVA was 3%. CVB was only detected in January, February, and July 2020, and the positivity rate of CVB was 1.8%. Echo was also detected in February, March, and April 2020, and the positivity rate was 1.3%. Only one HRV was detected in September 2020, and the positivity rate was 0.3%. AdV was only undetected in May, July, and November 2020 and in January, February, June, and July 2021. The positivity rate of AdV was 12.8%. Compared with surveillance viruses, MRVs were the major occurrence that appeared in this period, and the positivity rate was 59.4% (Table 1).

## 3. Discussion

Environmental surveillance by testing sewage for the evidence of pathogens has a long history of use in public health, particularly for poliovirus and the surveillance of antimicrobial resistance (AMR) in humans, such as monitoring resistance and the use of antimicrobial medicines, including AMR in the food chain and in the environment [17,18]. The role of environmental surveillance in supporting the Global Polio Eradication Initiative has already been acknowledged, and it can be used as a supplemental tool for the detection of pathogens circulating within the community [19]. Ahmed et al. [20] mentioned that the sewage surveillance system can pick up the vast majority of infected individuals with SARS-CoV-2 who do not present symptoms of the disease. Sewage provides near real-time data, as it constantly collects feces, urine, the sewage of households, and traces of sputum that can contain SARS-CoV-2 shed by infected individuals.

In this study, the routine viral surveillance of sewage throughout Taiwan was performed from January 2020 to July 2021. During this 19-month survey period, 397 raw sewage specimens were examined to detect the presence of viruses. The results showed that no poliovirus was found. Among the viruses, MRV (59.4%) and AdV (12.8%) predominated, followed by EV (5.3%). Compared with Lim et al. [13] in 2015, we found that MRV still had an increased positivity rate in sewage specimens in Taiwan. MRV belongs to the *Orthoreovirus* genus, *Reovirus* family, and *Spinareovirinae* subfamily, which commonly cause asymptomatic infections or mild respiratory tract illness and enteritis in infants and children. In previous studies, reoviruses were commonly found in environmental water sources, and human fecal contamination has been suggested as the source of the virus [21,22]. MRV may be a potential risk factor with public health implications. Even though MRV has frequently been identified in wastewater, no severe human case has been identified thus far in Taiwan.

The viral load in COVID-19 patients is still uncertain regarding their medical severity. However, Zhang and Wu reported 6 × 10^5^ viral genomes per mL of fecal material and 3 × 10^7^ viral particles in a single fecal sample. This may add another potential detection source of SARS-CoV-2 in communities [23].

Among the 11 sewage treatment plants selected in this research, 6 (54.5%) have a population density of ≥4910 inhabitants/km^2^, 3 (27.2%) have a population density of ≥1227 inhabitants/km^2^ and 2 have a population density of (18.2%) ≥307 inhabitants. Based on the sampling standards of the WHO poliovirus environment surveillance, the population covered by each sampling site should be between 1 and 3 × 10^5^ people. In this study, all the sampling sites exceeded the minimum of 1 × 10^5^ people required by the WHO. The population density of several sampling sites is close to 10,000 people per square kilometer. These 11 sewage sampling sites all cover more than 3 × 10^5^ residents, including the most densely populated areas in Taiwan (Figure 1). Therefore, we believe that the use of sewage sampling from sewage treatment plants should reflect the prevalence of the COVID-19 pandemic in Taiwan.

The poliovirus environmental sewage monitoring mechanism was established in Taiwan in 2012, and we have conducted additional SARS-CoV-2 surveillance since 2020. From January 2020 to July 2021, a total of 397 samples were analyzed. At the beginning of June 2021 (first sampling in June), two positive sewage specimen results were found at two sampling sites (A and C) in northern Taiwan. This is the first time that a positive result of SARS-CoV-2 nucleic acid was detected in environmental sewage in Taiwan. The numbers of confirmed COVID-19 cases in the regions where site A and site C occurred in June 2021 were 1748 and 3025, respectively. There were 4773 COVID-19 cases in the two regions, accounting for 76% of the 6298 confirmed cases in Taiwan at that time. Hence, it was reasonable to detect SARS-CoV-2 nucleic acid in those two sampling sites in June.

Potential factors affecting the detection of SARS-CoV-2 nucleic acid in wastewater are discussed below.

### 3.1. The Relationship between Sewage Treatment Plants and Population Density

Previous studies have mentioned that the concentration of SARS-CoV-2 in sewage is affected by the number of infected people in the catchment area [19,24]; however, there seems to be a discrepancy in the observed relationships between the concentrations of SARS-CoV-2 in sewage and infected people. La Rosa et al. [24] found that one of the positive results was obtained in a Milan sewage sample collected a few days after the first reported Italian case. In contrast, Medema et al. [19] and Randazzo et al. [25] obtained positive results from sewage samples before the first confirmed case in their studies.

This may have been, at least partly, because the number of confirmed cases did not completely reflect the actual prevalence of the infection at a sampling time point, which might be better correlated with the concentration of SARS-CoV-2 in the sewage. Another reason may be asymptomatic cases in the community. Furthermore, the catchment network of the sewage treatment area does not include sewage discharge lines for infection cases in the area. Even if it was detected once due to the presence of patients in the catchment area, it would no longer be detected if the patients were moved outside the area for hospitalization.

Sewage samples containing SARS-CoV-2 may be diluted so much that positive results cannot be detected unless they have accumulated a certain amount of the virus. Based on Taiwan COVID-19 surveillance data, a large number of infections in the site A and C catchment areas occurred after 16 May 2021, but it was not until June that the COVID-19 nucleic acid test was found to be positive. This finding is similar to the findings of La Rosa et al. [24]; a small number of positive cases cannot directly reflect the test results of sewage treatment plants, especially for sewage treatment plants designed for cities with a large population (>1 × 10^6^).

### 3.2. The Frequency of Sampling and Detection from Sewage Treatment Plants

Most of the previously published literature on sewage research set a very short time interval for research at the beginning of the 2020 pandemic [19,23,24]. The main reason is that the pandemic had already affected the aforementioned countries; therefore, they made quick inferences from the sewage results. Furthermore, the Netherlands plans to incorporate daily sewage surveillance into its national COVID-19 monitoring [26]. In contrast, the pandemic was well controlled in Taiwan in 2020, and the number of infections was small; thus, we conducted long-term monitoring of SARS-CoV-2 in sewage. In addition, according to the research data of Chin et al., the incubation time of TCID_50_ for SARS-CoV-2 at 22 °C is 7 days, declining to 1 day at 37 °C [12]. We determined the ambient temperatures of site A and site C in this study; the monthly average temperatures are 25.8–28.3 °C and 24.7–27.3 °C in May and June, respectively. If SARS-CoV-2 is discharged in the sewage treatment plant during this period, it is estimated that the TCID_50_ will decrease in 2–4 days. Therefore, if sampling takes more than 4 days to perform, it may not be possible to detect SARS-CoV-2. As temperature affects the detection of environmental SARS-CoV-2, the frequency of sampling should be shortened to once every 7 days or even shorter, especially in the summertime, which is usually over 35 °C in Taiwan.

### 3.3. The Impact of Rainfall

The monthly rainfall in May and June of 2021 in Taiwan was 119.5 and 316.5 mm for site A and 205 and 168 mm for site C, respectively (Figure 2). Both months recorded the highest rainfall since January 2021, at least for the past 6 months in 2021, and their rainfall was higher than the annual average of 117 and 141 mm at site A and site C, respectively.

The sewage samples in Taiwan contain not only water from household use but also outdoor rainwater. The sewage sewer covers about 80 [27] and 68% [28] of households in the Taipei (site A) and New Taipei City (site C) areas, respectively, and some household sewage discharges directly into a ditch, gutter, or river. Due to the higher rainfall in May and June this year, the concentration of SARS-CoV-2 in wastewater, it is inferred, may be diluted. This makes the Ct value of the two positive samples over 40. Haramoto et al. [29] also proposed the same inference that the sewer systems channeling domestic sewage and stormwater to wastewater treatment plants (e.g., combined or separate sewer) might influence the concentration of viral RNA in the wastewater reaching a wastewater treatment plant. Therefore, the early detection of SARS-CoV-2 may be limited in our setting.

### 3.4. Environmental SARS-CoV-2 Recovery

There is no standardized method for recovering SARS-CoV-2 from environmental samples, and most studies use the existing methods of their respective laboratories, including the method used by the WHO’s routine polio monitoring efforts (PEG precipitation) [13,23,24]. Ahmed et al. [20] noted that the electronegative membranes had a higher recovery rate (60–65%); the recovery rate of filtration membrane centrifugation was 28–56%, the recovery rate of PEG precipitation was 44%, and the recovery rate of ultracentrifugation was 33.5%. The advantages of various methods, such as electronegative membranes, include short operation times and the ability for them to be conducted in the field at the same time, but the disadvantage is that clogging can easily occur. The filtration membrane centrifugation method also has a short operating time, but the disadvantage is that it is expensive. The PEG precipitation method is relatively cheap, but the steps are cumbersome and time-consuming. Ultracentrifugation can concentrate the solid phase and liquid phase, but it is time-consuming and can only be performed in a small volume.

Therefore, most studies recommend that each country adopts its own suitable methods to concentrate sewage. To avoid increasing the burden of labor, we adopted the PEG precipitation method. At the same time, because SARS-CoV-2 is a membrane virus, we revised some of the processing procedures without adding chloroform to destroy the virus. In addition, we added two filtration steps for direct use in automated nucleic acid extraction and analysis systems.

Considering the low concentrations of SARS-CoV-2 in sewage, further studies to explore a more effective method for concentration and RNA extraction are recommended.

### 3.5. Detection Method

Conventional real-time molecular diagnostic methods include several manual steps, such as RNA/DNA extraction, master mix preparation and RT–PCR, as well as the interpretation of results. These steps are labor intensive and time-consuming. We adopted a fully automated system that allows for the handling of large numbers of samples and significantly reduces the hands-on time. In addition, it is easy to train personnel who are not familiar with molecular diagnostic assays. However, some of the limitations of automated tests are that specific specimens cannot be placed directly on the instrument, and these tests may be too expensive for most laboratories [30].

Taiwan has successfully used effective contact tracing management and simultaneously used a series of public health measures to monitor people who need to be quarantined [31]. At the same time, a real-name text message registration system was used to effectively prevent the pandemic. Therefore, compared to other countries in Asia, the number of patients infected with COVID-19 in Taiwan is still limited.

At present, there is not yet sufficient evidence to recommend environmental surveillance as a standard approach for COVID-19 surveillance [32]. Factors such as collection at different time points, sample concentration, and sewage collection and treatment in a large area may affect the determination of early warning signs. Therefore, expanding the establishment of sampling sites and shortening the collection time interval may help to establish a more comprehensive SARS-CoV-2 surveillance system. However, our results from the environmental sewage surveillance system can still be used as an auxiliary reference for monitoring the trends of COVID-19 in Taiwan.

## 4. Materials and Methods

### 4.1. Sewage Collection

From January 2020 to July 2021, specimens of untreated sewage were collected in northern, central, southern, and eastern Taiwan every two weeks at monthly intervals from 11 (sites A to M) representative sewage treatment plants (Figure 1) to treat maximum volumes of 500,000; 240,000; 42,000; 166,250; 40,000; 132,000; 75,000; 800,000; 20,000; 50,000; and 100,000 cubic meters per day, respectively. A one-liter sample was taken from the raw sewage inlet of the sewage treatment plant and sent to the Taiwan CDC for analysis. The sample was kept at 4 °C during transportation before handling.

### 4.2. Concentration of Sewage Specimens

For the treatment of sewage samples, we adopted the two-phase separation method. This method is mainly performed in accordance with the World Health Organization (WHO) Poliovirus Transmission Environmental Monitoring Guidelines and the WHO Polio Laboratory Manual Guidelines. The main function of polyethylene glycol (PEG) precipitation [33] is to concentrate viruses in sewage. The related literature also suggests that using PEG to precipitate from sewage and surface water samples is effective for concentrating viruses with membranes [34,35]. For the concentration of poliovirus and SARS-CoV-2, we used the same wastewater sample to separate two parts for treatment (Figure 3).

For routine poliovirus monitoring projects, 500-milliliter samples were centrifuged at 1000× *g* at 4 °C for 10 min, and the precipitate was stored at 4 °C. The supernatant was transferred to a new bottle, and the pH was adjusted to neutral (7.0–7.5). Then, 39.5 mL of 22% dextran, 287 mL of 29% PEG, and 35 mL of 5 N NaCl were added to the supernatant and shaken for 1 h. The supernatant was transferred to a sterile conical separatory funnel and allowed to stand overnight. Then, the entire lower layer and middle-phase liquid were collected and mixed with the first centrifugal sediment; then, a 20% volume of chloroform was added to the mixture volume, the mixture was centrifuged at 10,000× *g* for 30 min, and the upper water phase was removed and dissolved with 0.5 mL of PBS. A subsequent analysis was performed after the RNA extraction procedure. The upper water phase was collected, and antibiotic–antimycotic solution (100X) was added. Then, 200 μL of concentrated specimen was inoculated into cell lines.

The protocol was partially adjusted for SARS-CoV-2 based on a previous step. The chloroform treatment at this stage of the WHO protocol was omitted to preserve the integrity of the enveloped viruses that were the object of this study. In brief, 500 mL of sample was centrifuged at 1000× *g* for 10 min at 4 °C, and the precipitate was stored at 4 °C. The supernatant was transferred to a new bottle, and the pH was adjusted to neutral (7.0–7.5). Then, 39.5 mL of 22% dextran, 287 mL of 29% PEG, and 35 mL of 5 N NaCl were added to the supernatant, which was then shaken for 1 h. The supernatant was transferred to a sterile conical separatory funnel and allowed to stand overnight. Then, the entire lower layer and middle-phase liquid were collected and mixed with the first centrifuged sediment. The mixture was centrifuged at 10,000× *g* for 30 min [36]. After the supernatant was removed, it was dissolved in 1.5 mL of PBS and filtered with 1.2-micrometer and 0.45-micrometer filter membranes. Then, the filtration specimens were used in the automated molecular assay.

### 4.3. Routine Polio Surveillance System for Inoculation and Immunofluorescent Staining (IFA)

According to Lim et al. [13], a previous study with modifications, human rhabdomyosarcoma (RD) cells (ATCC CCL-136™) and recombinant murine L20B cells were used for the isolation of environmental viruses. Concentrated specimens (200 μL) were inoculated into each cell culture tube, incubated at 35 °C, and examined for cytopathic effects (CPE) daily. On day 7, negative CPE tubes were blindly passed to a new cell culture tube and observed daily until day 14. Positive CPE specimens were stained with a pan-enterovirus (PanEV) blend antibody and other virus-screening antibodies.

### 4.4. Routine Polio Surveillance System for RNA/DNA Extraction

Viral RNA was extracted from 0.3 mL of the concentration sample using the TANBead Maelstrom 4800 automated Nucleic Acid Extractor (TANBead, Taipei, Taiwan) according to the manufacturer’s instructions.

### 4.5. Routine Polio Surveillance System for Molecular Analysis

The CODEHOP PCR method developed by Nix et al. [37] with modifications for PanEV-positive untypable specimens with a reverse transcription semi-nested PCR (RT-snPCR) and sequencing was used.

### 4.6. SARS-CoV-2 Molecular Assay

#### 4.6.1. SARS-CoV-2 In-House Real-Time PCR Assay

From January to March 2020, as there was no approved rapid commercial reagent for EUA, we started using an in-house real-time PCR at the beginning of COVID-19 in Taiwan. In the absence of a standardized method for SARS-CoV-2 detection in environmental samples, RNAs were tested for different assays. Viral RNA was extracted from 0.3 mL of the sample using the TANBead Maelstrom 4800 automated Nucleic Acid Extractor (TANBead, Taipei, Taiwan) according to the manufacturer’s instructions. Quantification RT-PCR (qRT–PCR) was performed using the TaqMan^®^ Fast virus 1-Step Master Mix reagent (Thermo Fisher Scientific, Waltham, MA, USA) with the previously published primer/probe. Briefly, the 25-microliter qRT–PCR mixture contained TaqMan^®^ Fast virus 1-step Master Mix, 400 nM each of forward and reverse primer, 200 nM TaqMan probe, and 5 μL of extracted RNA (or water for the no-template controls). Positive detection of SARS-CoV-2 using the Taiwan CDC protocol was considered when a sample tested positive for the E, RdRp, and N genes. Samples were considered SARS-CoV-2 negative if they tested negative for the E and RdRp genes or negative for the RdRp gene but positive for the E gene [30,38].

#### 4.6.2. Automatic Extraction and Molecular Detection Assay for SARS-CoV-2

To avoid PCR contamination, we used the Xpert Xpress SARS-CoV-2 test [39] (Cepheid, Sunnyvale, CA, USA), which is the first POC test approved by the US FDA EUA (March 2020) and also approved by the European CE IVD. The systems are intended for single-use disposable cartridges. The GeneXpert Instrument Systems automate and integrate sample preparation, nucleic acid extraction and amplification, and detect the two target sequences (N2 and E) using real-time RT PCR assays. The results are interpreted automatically by the GeneXpert System. The detection sensitivity is 33.3–83.3 copies per reaction, and there was no potential unintended cross reactivity with thirty-two different organisms. Starting in April 2020, we retrospectively confirmed the experimental results from January to March 2020. Briefly, 0.3 mL of the filtered product was directly placed into the Xpert Xpress SARS-CoV-2 cassette and analyzed according to the original manufacturer’s recommendations.

## Figures and Tables

**Figure 1 pathogens-10-01611-f001:**
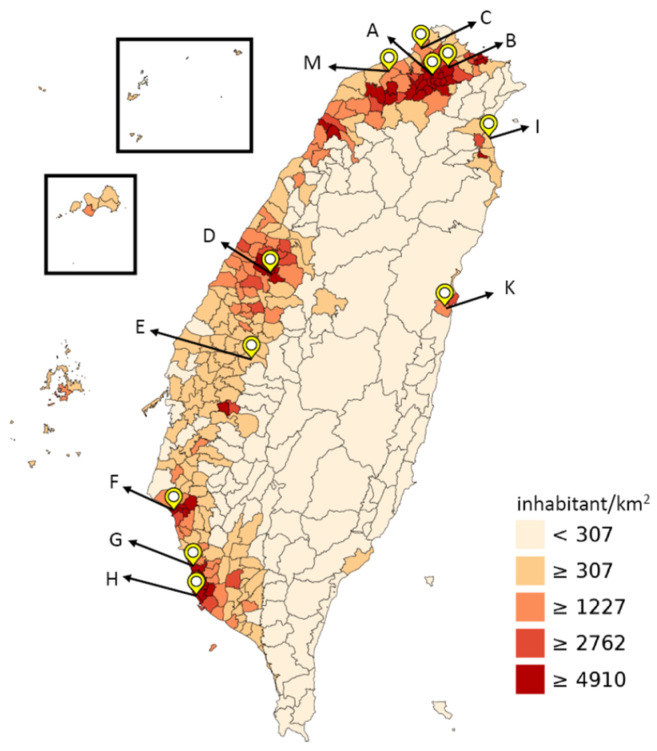
Sewage collection sites and county population densities in Taiwan. Site A, Taipei (Dihua); site B, Taipei (Neihu); site C, New Taipei City; site D, Taichung; site E, Yunlin; site F, Tainan; Site G, Kaohsiung (Nanzi); site H, Kaohsiung (Qijin); site I, Yilan; site K, Hualien; site M, Taoyuan. Modified from http://commons.wikimedia.org/wiki/File%3APopulation_density_of_Taiwan_by_district.svg (accessed on 24 September 2021) under a Creative Commons Attribution Share-Alike 3.0 (CC BY-SA 3.0) license, with permission from Wikimedia Commons, 2013.

**Figure 2 pathogens-10-01611-f002:**
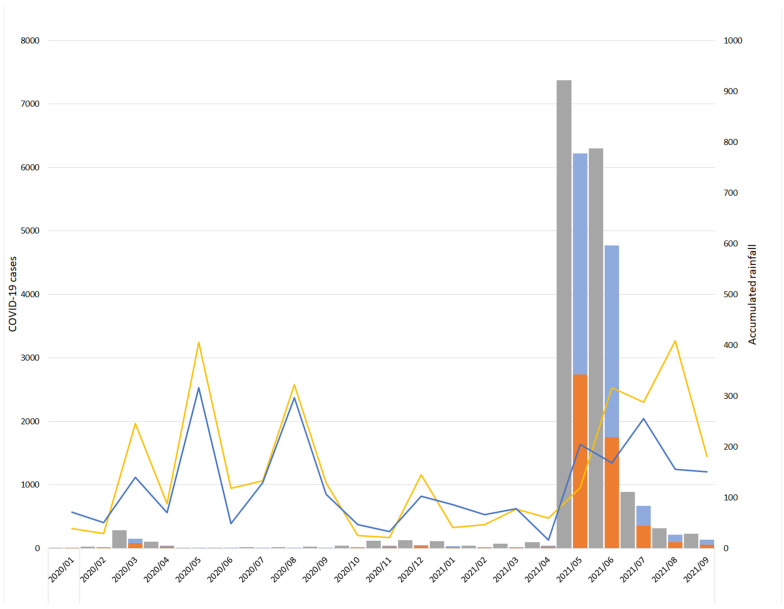
Trend of COVID-19 in total and two sampling sites cases between January 2020 and September 2021 compared with accumulated rainfall in Taiwan.

**Figure 3 pathogens-10-01611-f003:**
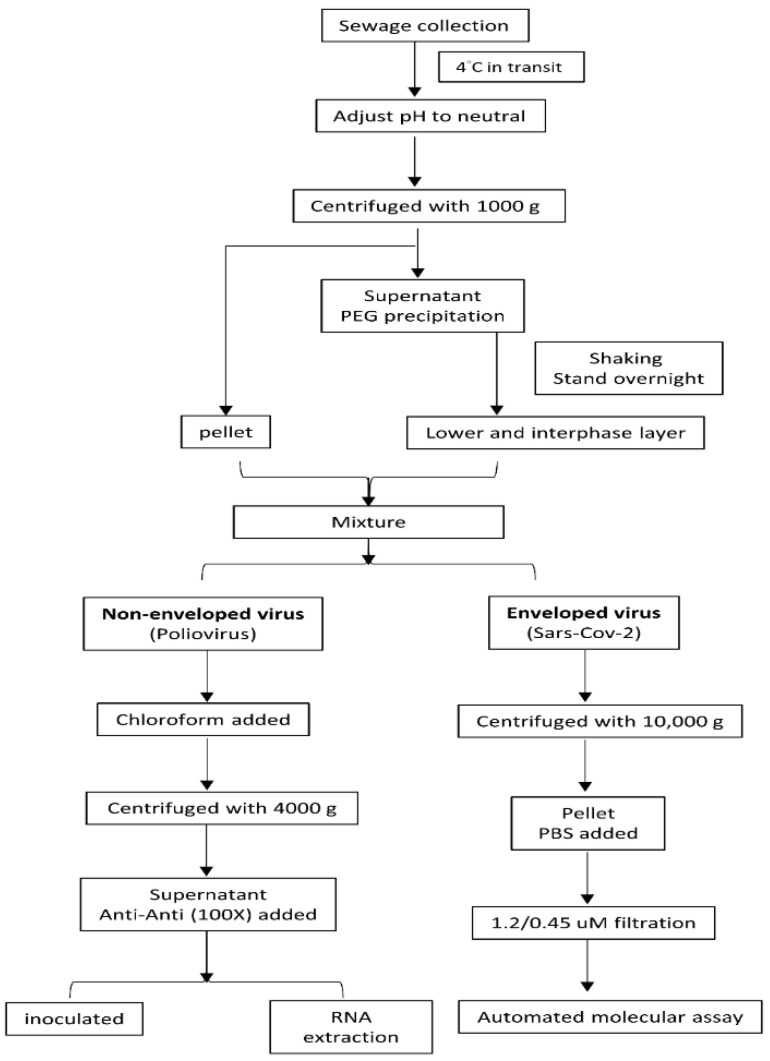
Flow diagram of the sample processing for poliovirus surveillance and SARS-CoV-2 concentration and detection in Taiwan CDC.

**Table 1 pathogens-10-01611-t001:** Distribution of virus isolated from wastewater samples in Taiwan (January 2020 to July 2021).

Year	2020												2021						
Month	Jan	Feb	Mar	Apr	May	Jun	Jul	Aug	Sep	Oct	Nov	Dec	Jan	Feb	Mar	Apr	May	Jun	Jul
Total specimens tested	11	22	22	22	22	22	22	22	22	22	22	22	23	11	22	22	22	22	22
Poliovirus	0	0	0	0	0	0	0	0	0	0	0	0	0	0	0	0	0	0	0
EV *	1	5	2	3	3	0	0	1	0	2	0	0	1	0	2	1	0	0	0
CVA *	1	3	1	0	1	0	0	4	2	0	0	0	0	0	0	0	0	0	0
CVB *	3	3	0	0	0	0	1	0	0	0	0	0	0	0	0	0	0	0	0
Echo *	0	2	2	1	0	0	0	0	0	0	0	0	0	0	0	0	0	0	0
HRV *	0	0	0	0	0	0	0	0	1	0	0	0	0	0	0	0	0	0	0
AdV *	7	6	5	3	0	1	0	3	8	2	0	2	0	0	6	4	4	0	0
MRV *	10	18	19	8	6	17	19	17	12	16	5	7	13	9	9	11	14	10	16
SARS-CoV-2	0	0	0	0	0	0	0	0	0	0	0	0	0	0	0	0	0	2	0

* Enterovirus (EV), coxsackievirus A (CVA), coxsackievirus b (CVB), echovirus (Echo), human rhinovirus (HRV), adenoviruses (AdV), and Mammalian orthoreovirus (MRV).

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
