# Peer review of "Surveillance of SARS-CoV-2 in Sewage Treatment Plants between January 2020 and July 2021 in Taiwan"

_pathogens, 2021, doi:10.3390/pathogens10121611_

Round 1
Reviewer 1 Report
In the article “Surveillance of SARS-CoV-2 in Sewage Treatment Plants between Jan. 2020 and Jul. 2021 in Taiwan”, the authors provide information about the levels of different viruses in the wastewater of several sites. After reading the article, the reviewer is still not convinced that the measurement of Sars-Cov-2 from sewage provides any benefit for the management of outbreaks. Furthermore, the title is also misleading, because there is very little actual data about Sars-Cov-2 in the article, therefore the conclusions are not supported by results. The reviewer can’t recommend this article for publication in a journal with an IF ≈ 4.
Reviewer 2 Report
Major concern is the lack of proper positive and negative controls for qRT-PCR assay reporting SARS-COV-2 RNA. As the reported values as very high (Ct 41, above normal threshold cut-off), this could be an artifact of the assay or background rather than true positive SARS-CoV-2 results.
Please, find additional comments below:
Line 58: Figure 1 needs to be cited earlier in the paragraph.
Figure 1: Accumulated rainfall and site A accumulated rainfall is shown in the figure but data is not explained in results section. Authors talk about rainfall in discussion, so Figure needs to be cited there as well.
Site A and C data is shown in figure 1 but location of sites is shown in Figure 2 and described only in Discussion section. To improve readability of the manuscript, sites location needs to be introduced early in the manuscript.
Line 94: Additional citations are needed to support this statement as the article selected here on fecal-oral transmission is based on SARS from 2002 outbreak not SARS-CoV-2.
Line 126: Any study reporting SARS-CoV-2 or SARS from earlier outbreaks in wastewater in other countries? Need to be introduced and cited in introduction.
Line 132: what is site M? Needs to be defined here or figure 2 presented first.
Line 135: Ct 41.1 and 41.8 referring to qRT-PCR cycle. Normal cycle cut-off is 40. This could be due to background and not positive SARS-COV-2. Proper controls need to be included in the study and presented in the manuscript to validate sensitivity and specificity of the assay.
Lines 267-282: cite Figure 1
Line 278: What controls were used to confirm the statement that ct 41 is due to high rainfalls in the sewage? Such higher ct value could be just the background.
Reviewer 3 Report
The research work done by Jyh-Yuan Yang and co-workers “Surveillance of SARS-CoV-2 in Sewage Treatment Plants between Jan. 2020 and Jul. 2021 in Taiwan” detailed a good research report for the understanding of SARS-CoV-2 in Sewage Treatment Plants. Authors successfully documented efficient, simple and straightforward approach for the surveillance of SARS-CoV-2 in Sewage Treatment Plants. Though the similar approaches reported by few other research groups, they didn’t utilize the surveillance of SARS-CoV-2 in sewage treatment plants approach as authors documented, where it is important in studying and analyzing the SARS-CoV-2 virus. Given the importance of practicality for this work, I recommend the publication of this manuscript in the “pathogens” to accept.
Round 2
Reviewer 2 Report
New version of the manuscript (v2) is identical to the previous version and does not include any of the changes described by the author in the response letter.
Author Response
Thank you for your comments and we revise it point-by-point. Please see the attachment.
